# Effect of antibiotic medicines availability on adherence to standard treatment guidelines among hospitalized adult patients in southern Malawi

**Francis Kachidza Chiumia**[1], **Adamson Sinjani Muula**[2], **Frider Chimimba**[1], **Happy Magwaza Nyirongo**[1], **Elizabeth Kampira**[3], **Felix Khuluza**[1] *

1 Department of Pharmacy, School of Life Sciences and Allied Health Professions, Kamuzu University of Health Sciences, Blantyre, Malawi, 2 Department of Community and Environmental Health, School of Global and Public Health, Kamuzu University of Health Sciences, Blantyre, Malawi, 3 Department of Medical Laboratory Sciences, School of Life Sciences and Allied Health Professions, Kamuzu University of Health Sciences, Blantyre, Malawi

* fkhuluza@kuhes.ac.mw

**Data Availability Statement:** All relevant data are within the manuscript and its Supporting Information files.

## Abstract

### Background

Antibiotic resistance is a global public health problem. High and inappropriate use of antibiotic therapy exacerbate the risk of antibiotic resistance. We assessed the effect of availability of antibiotic medicines on adherence to standard treatment guidelines among hospitalized adult patients in Southern Malawi.

### Methods

A cross-sectional study was done to assess the availability of 16 antibiotics among the first-line recommended treatments for common bacterial infections in Malawi. Data for up to six-month duration was extracted from stock card records in Machinga and Nsanje District Hospitals and Zomba Central Hospital. This was complemented by a retrospective review of 322 patient management files from medical wards to assess adherence to the Malawi Standard Treatment Guidelines (MSTG). Investigators abstracted data such as patient demographics, diagnoses, and prescribed therapy using a data collection form that resulted in analyzing 304 patient files. Data was entered into Microsoft excel and analyzed using STATA 14.1. Point availability, stock-out duration and adherence to treatment guidelines were presented in terms of frequencies and percentages. Chi-square test or Fisher's exact test was applied to assess the association between variables and adherence to treatment guidelines.

### Results

Point availability of antibiotics was 81.5%, 87.7%, and 42.8% for Zomba Central, Machinga and Nsanje District Hospitals respectively. Over a period of six months, 12.5% of antibiotic medicines were stocked out for at least one day at Zomba (Median stock out days = 0, (IQR

**Funding:** FK is the recipient of EDCTP 2 grant number TMA2019CDF-2768_COPSMEDS. https://edctpalumninetwork.org/fe/profiles/view/102fecdc-bc9f-452a-934b-d9a8c3c7079e This study was part of the European and Developing Countries Clinical Trials Partnership 2 (EDCTP2) programme supported by the European Union (under grant number TMA2019CDF-2768 COPSMEDS). The funders had no role in the study design, data collection and analysis, decision to publish, or preparation of the manuscript.

**Competing interests:** The authors have declared that no competing interests exist.

0–0 days), while 64.3% were stocked out at Machinga (Median stock out days = 21, IQR 0–31 days) and 85.7% were stocked out at Nsanje District Hospital (Median stock out days = 66.5, IQR 18–113 days). Overall, adherence to MSTG was 79.6%, (95% CI, 73.3–84.9%). By facilities, adherence to guidelines at Zomba Central Hospital was 95.9% (95% CI, 89.7–98.9%) while at Nsanje and Machinga District Hospitals was 73.2% (95% CI, 59.7–84.2%) and 54.2% (95% CI, 39.2–68.6%) respectively. Adherence to treatment guidelines was associated with health facility, presence of laboratory test results, antibiotic spectrum, and WHO-AWaRe category of the medicine, p<0.005. Adherence was lower for antibiotics that were stocked out than antibiotics that were not stocked out during the study period (63.8%, 95% CI 48.5–77.3% vs 84.4%, 95% CI 77.7–89.8%), p< 0.002.

## Conclusion

We found unstable availability of antibiotic medicines in hospitals which might contribute to the sub-optimal adherence to standard treatment guidelines. This is a setback to efforts aimed at curbing antibiotic resistance in Malawi.

## Background

Bacterial infections are amongst the leading causes of mortality globally [1]. Nearly 13% of deaths around the world are attributed to bacterial infections [2]. There are concerns of increasing incidences of antibiotic resistance due to the haphazard use of antibiotic therapy [3, 4]. In 2019, it was estimated that about five million deaths were associated with antibiotic resistance globally. Deaths associated with antibiotic resistance were highest in sub-Saharan Africa, with 98.9 deaths per 100,000 while in high-income countries the rate was 55.7 deaths per 100,000 [5]. The high burden of antibiotic resistance in low and middle-income countries (LMICs) is attributed to both high and inappropriate use of antibiotics [6]. In addition, an unstable supply of essential medicines in LMICs influences the inappropriate selection of antibiotic therapy [7].

A study conducted across 20 LMICs observed that more than 60% of out-patient pediatric patients were prescribed antibiotics, but their availability was as low as 40% in the majority of the countries [8]. In Malawi, the availability of essential medicines is limited as only about 50% of essential medicines were constantly available in public health facilities in 2017 [9]. Medicine availability may, however, vary depending on the type of medicine, health facility and season. For instance, the availability (in public health facilities) of adult formulation of amoxicillin, a commonly prescribed oral beta-lactam antibiotic [10, 11] was 60% in 2015 [12], while in 2017 another study revealed a higher availability of 100% [9]. Data for the two cross sectional studies was collected at different times. In addition, the latter study focused on public health facilities located in Blantyre and Lilongwe cities, while the other study focused on Blantyre city as well as three other surrounding rural districts.

Constant availability of medicines is one of the key indicators of a country's performance towards universal health coverage [13]. Essential medicine lists provide guidance on the prioritized medicines that are supposed to be in sufficient supply at all times in order to meet the healthcare needs of a country [14]. On the other hand, standard treatment guidelines are necessary to ensure the rational use of these medicines [15]. Adherence to treatment guidelines significantly reduces mortality and morbidity [16]. Furthermore, it improves cost-effectiveness and medicine supply chain efficiency [15].

Implementation of the essential medicine list concept in Malawi has been affected by several challenges such as higher medicine prices than what most hospitals can afford, inefficient procurement systems, and poor healthcare infrastructure [17]. The latest edition of the Malawi Essential Medicine List (MEML) was published in 2015 and is incorporated with the Malawi Standard Treatment Guidelines (MSTG) [18]. Due to the high prevalence of infectious diseases, the MEML gives much attention to antimicrobial agents, of which the majority are antibiotics [17]. Thus, as is the case in other LMICs, most prescriptions in Malawi are antibiotic therapy [19–21]. The Malawi antimicrobial resistance strategy was established in 2017 and aimed at achieving 100% optimization of the use of antibiotic therapy by 2022 [22]. In this study, we evaluated the availability of selected antibiotic medicines and its influence on adherence to national guidelines (MSTG) for prescribing antibiotics among hospitalized adult patients.

## Methods

### Study design and setting

A cross sectional study was conducted to collect data on point availability of antibiotic medicines in February 2022. This was supplemented by data on medicine stock-outs in the past six months. To assess the adherence of therapy to national standard treatment guidelines, we retrospectively reviewed records for patients admitted to medical wards. The study was conducted in two secondary-level health facilities (district hospitals) in the Machinga and Nsanje districts and a tertiary-level health facility (central hospital) in Zomba district. The districts were randomly selected among 13 districts in southern Malawi using RAND function in Microsoft Excel.

### Criteria for selection of antibiotic medicines

We purposively selected first-line antibiotics for common bacterial infections as stated in the 2015 Malawi Standard Treatment Guidelines (MSTG) which was the latest version at the time of the study [18]. Common bacterial infections in Malawi include: sepsis, respiratory tract infections such as pneumonia, sinusitis, and bronchitis, HIV and AIDS complications such as meningitis, urinary tract infections such as cystitis and urethritis, cellulitis and other skin conditions, and sexually transmitted diseases such as syphilis and genital ulcers [23–29]. In this study, we included the first line antibiotic therapy for these common conditions as recommended in the MSTG. **S1 Table** provides more details on the treatment protocols for the conditions. Thus, we included gentamicin, ceftriaxone, erythromycin, azithromycin, clarithromycin, metronidazole, amoxicillin, cloxacillin, flucloxacillin, benzylpenicillin, benzathine benzylpenicillin, ciprofloxacin, nalidixic acid, co-trimoxazole, doxycycline and meropenem. The selected antibiotics represent 45.7% of all antibiotics in adult formulations (N = 35) in the MEML. Unlike the other antibiotics, meropenem and clarithromycin are mostly found in tertiary level hospitals, as they are reserved for rare and life-threatening conditions caused by multi-drug resistant bacteria [30]. As such, the availability of meropenem and clarithromycin was only assessed at tertiary hospital and not the secondary (district) level hospitals. As for both benzathine benzylpenicillin and benzylpenicillin, they were included as they are used for two different conditions. Benzathine benzylpenicillin is mainly used as first-line treatment for sexually transmitted diseases in Malawi while benzylpenicillin is for routine infections. This is because the addition of benzathine makes the combination more long-acting as such ideal for sexually transmitted diseases unlike benzylpenicillin which needs four time administration.

## Study population and sampling

A total of 322 patient records were randomly sampled across the facilities. The sample size was determined using a single population proportion formula: n = p(1-p) *$Z^2$/$d^2$ [31]. We estimated p as 30%, 0.05 margin of error (E) and 1.96 Z value corresponding to 0.05 significance level. The inclusion criteria were hospitalized patients with aged ≥ 18 years old, who were given antibiotics as the main treatment. Medical records with missing information such as medication charts and demographic information were excluded from the study.

## Data collection

FKC, HMN, and FK reviewed stock cards in medicine storage facilities to collect data on the availability of antibiotic medicines on the day of data collection (point availability) and over the past six months using a data collection form (S2 Table). This data included stock remaining on the shelf and the number of stocked-out days within the six-month duration for each medicine type. Stocked-out days were obtained by counting the number of days the product was not available for use in the stock card from the day the balance was zero to the day when they received a new supply. A new supply would be received from either Central Medical Stores Trust, private pharmaceutical suppliers, or other health facilities in line with the Malawi Health Commodity Logistics Management system Standard Operating Procedures Manual [32]. A stock card (also called an inventory control card) is a stock-keeping record that is used for recording the inflow and outflow of pharmaceuticals in Malawi [33]. Generally, stock cards are kept in the Pharmacy for seven years before they are sent for archiving at the central office, as such all information for this study was available in the facilities. FKC, HMN, and FK further abstracted demographics and clinical data such as patient diagnosis and prescribed therapy from case management files of eligible patients. Antibiotic treatment was compared with treatment protocols in the 2015 edition of Malawi Standard Treatment Guidelines for the specified diagnosis, to assess for adherence to treatment guidelines.

## Statistical analysis

All data were entered into Microsoft Excel and analyzed in STATA version 14.1. Availability of medicines was described in terms of percentages of antibiotic medicines that were not available on the day of data collection or at least one day in the past six months. Furthermore, stock-out days for each medicine type were described in terms of frequencies. Antibiotic regimens that were not in line with the MSTG were quantified and Chi-square test was used to assess the association between variables and adherence to the guidelines. Only in cases where the cell number was ≤ 5, Fisher's exact test was applied.

# Results

## Classification of antibiotic medicines

Availability of medicines was assessed among 16 antibiotics. These were five penicillin antibiotics, three macrolides, two quinolones, one aminoglycoside, one cephalosporin, one nitroimidazole, one tetracycline, one carbapenem and one fixed dose combination of sulfonamide and trimethoprim. According to the WHO Access, Watch and Reserve (AWaRe) classification [34], 56% (9/16) were access antibiotics, 38% (6/16) were watch antibiotics, and one antibiotic was unclassified. Anatomical, chemical, and therapeutic categories are shown in Table 1. According to Malawi Essential Medicines List (MEML), 87.5% (14/16) of the antibiotic medicines were classified as vital medicines, 12.5% (2/16) were essential medicines and none of the antibiotics was classified as non-essential medicine. In terms of level of access, all the antibiotic

**Table 1. Classifications of antibiotic medicines.**

| Antibiotic name (generic) | Dosage form | Antibiotic class | ATC code[1] | Spectrum of activity | AWARE category | MEML category[2] |
|---|---|---|---|---|---|---|
| Gentamycin | Injectable | Aminoglycoside | J01GB03 | Broad | Access | HVA |
| Ceftriaxone | Injectable | Cephalosporin | J01DD04 | Broad | Watch | DVA |
| Erythromycin | Solid (Tabs or caps) | Macrolide | J01FA01 | Broad | Watch | HVA |
| Azithromycin | Solid (Tabs or caps) | Macrolide | J01FA10 | Broad | Watch | DEA |
| Clarithromycin | Solid (Tabs or caps) | Macrolide | J01FA09 | Broad | Watch | CVA |
| Metronidazole | Solid or injectable | Nitroimidazole | J01XD01 or J01AB01 | Broad | Access | HVA |
| Amoxicillin | Solid (Tabs or caps) | Penicillin | J01CA04 | Broad | Access | HVA |
| Cloxacillin | Solid (Tabs or caps) | Penicillin | J01CF02 | Broad | Access | DEA |
| Flucloxacillin | Solid (Tabs or caps) | Penicillin | J01CF05 | Narrow | Access | DVA |
| Benzyl penicillin | Injectable | Penicillin | J01CE01 | Narrow | Access | HVA |
| Benzanthine penicillin | Injectable | Penicillin | J01CE08 | Narrow | Access | HVA |
| Ciprofloxacin | Solid (Tabs or caps) | Quinolone | J01MA02 | Broad | Watch | DVB |
| Nalidixic acid | Solid (Tabs or caps) | Quinolone | J01MB02 | Narrow | Unclassified | DVA |
| Cotrimoxazole | Solid (Tabs or caps) | Sulfonamide-trimethoprim-combination | J01EE01 | Broad | Access | HVA |
| Doxycycline | Solid (Tabs or caps) | Tetracycline | J01AA02 | Broad | Access | HVA |
| Meropenem | Injectable | Carbapenem | J01DH02 | Broad | Watch | CVA |

[1]ATC = Anatomical, therapeutic and chemical classification.

[2]The Malawi Essential Medicines List (MEML) of 2015 specifies the level of health institution at which the medicine is normally permitted for use: H = at health centre, district hospital and central hospital levels; D = at district hospital and central hospital levels only; C = at central hospital level only; N = level of use not specified. The 'therapeutical priority' code categorizes medicines based on therapeutic importance of each medicine by the use of: V = vital medicines which are potentially life-saving, of major public health relevance and having significant withdraw side-effects; E = essential medicines which are effective against less severe, but nonetheless significant forms of illness; N = non-essential medicines which are used for minor self-limiting illness and are often of questionable efficacy. The third categorization of 'procurement system' has two codes: 'A' = medicines required by a large number of patients as such to be routinely procured and stocked by CMST; and 'B' = medicines required for a limited number of patients and not routinely stocked by CMST).

medicines were required to be available at both district and central hospitals except for meropenem and clarithromycin which were designated to be available at central hospitals only.

## Patient characteristics

From the 322 patient files, 18 files were excluded due to missing data; thus, a final total of 304 patient files were analyzed. Of the 304 patients, 48% were female with a mean age of 45.9 years, range (18–96 years), while 52% were male with a mean age of 46.1 years, range (18–92 years). Antibiotics were used for treatment in 71.4% (n = 217) of cases while they were used for prophylaxis in 28.6% (n = 87) of the cases (**Table 2**). No bacterial culture or antibiotic sensitivity test was conducted to aid in the diagnosis and selection of antibiotics respectively. Full blood count (FBC) was done in 56.6% (n = 172) of the patients. Only broad-spectrum antibiotics were used in 71.7% (n = 218) of patients while narrow-spectrum antibiotics were used in 7.6% (n = 23) of patients. The remaining 20.7% (n = 63) received a combination of broad and

**Table 2. Patient's characteristics and reasons for receiving antibiotic therapy.**

| Variable | Characteristic | Antibiotics used for treatment n (%) | Antibiotics used for prophylaxis n (%) |
|---|---|---|---|
| Age | Years | 43 (30–62)[1] | 46 (35–60)[1] |
| Sex | Male | 113 (52.1) | 45 (51.7) |
| | Female | 104 (47.9) | 42 (48.3) |
| Facility | ZA CTL | 99 (71.7) | 39 (28.28) |
| | MGH DHO | 55 (65.5) | 29 (34.5) |
| | NE DHO | 63 (76.8) | 19 (23.2) |
| Antibiotic types given | Number per prescription | 1 (1–2)[1] | 1 (1–2)[1] |
| Antibiotic spectrum | Broad spectrum antibiotics | 159 (72.9) | 50 (27.1) |
| | Narrow spectrum antibiotics | 10 (43.5) | 13 (56.5) |
| | Broad + Narrow spectrum antibiotics | 48 (76.2) | 15 (23.8) |
| WHO AWaRe group | Access group | 49 (65.3) | 26 (34.7) |
| | Watch group | 101 (71.63) | 40 (28.4) |
| | Access + Watch group | 67 (76.1) | 21 (23.9) |
| Full Blood count | Done | 126 (73.3) | 46 (26.7) |
| | Not done | 91 (68.9) | 41 (31.1) |
| Bacterial culture | Done | 0 (0) | 0 (0) |
| | Not done | 217 (71.4) | 87 (28.6) |
| Sensitivity test | Done | 0 (0) | 0 (0) |
| | Not done | 217 (71.4) | 87 (28.6) |

[1]Presented as Median (Interquartile range)

narrow spectrum antibiotics. Among those who received broad-spectrum antibiotics only, 72.9% (n = 159) were for treatment whilst 27.1% (n = 50) were for prophylaxis. Of those who received narrow spectrum antibiotics, 43.5% (n = 10) were for treatment while 56.5% (n = 13) were for prophylaxis. By AWARE classification, 24.6% (n = 75) received access antibiotics, 46.4% (n = 141) watch antibiotics and 29% (n = 88) received a combination of access and watch antibiotics. Among those who received access antibiotics, 65.3% (n = 49) were for treatment while 34.7% (n = 26) were for prophylaxis. For those who received watch antibiotics, 71.6% (n = 101) were for treatment while 28.4% (n = 40) were for prophylaxis.

## Medicine availability

There was a significant variation in antibiotic availability among facilities, p< 0.035. On the days of data collection, 81.5% (13/16) of the antibiotics were available at Zomba Central Hospital, 87.7% (12/14) at Machinga District Hospital, and 42.8% (6/14) were available at Nsanje District Hospital (**Fig 1**). Medicines that were not available at Zomba were erythromycin, clarithromycin, and cloxacillin. For Machinga, medicines that were not available were cloxacillin and nalidixic acid while for Nsanje, it was gentamycin, erythromycin, azithromycin, metronidazole, amoxicillin, cloxacillin, ciprofloxacin, and nalidixic acid. Zomba Central Hospital had the lowest rate of stock-outs in the past six months from the day of data collection. Only 12.5% (2/16) of the medicines were stocked out for at least one day. On the other hand, 64.3% (9/14) and 85.7% (12/14) of the medicines were stocked out at Machinga and Nsanje district hospitals respectively.

## Stock out duration

Stock out duration was highest at Nsanje, followed by Machinga and Zomba. Median stock out days were 66.5 days (IQR 18–113 days), 21 days (IQR 0–31 days), and 0 days (IQR 0–0

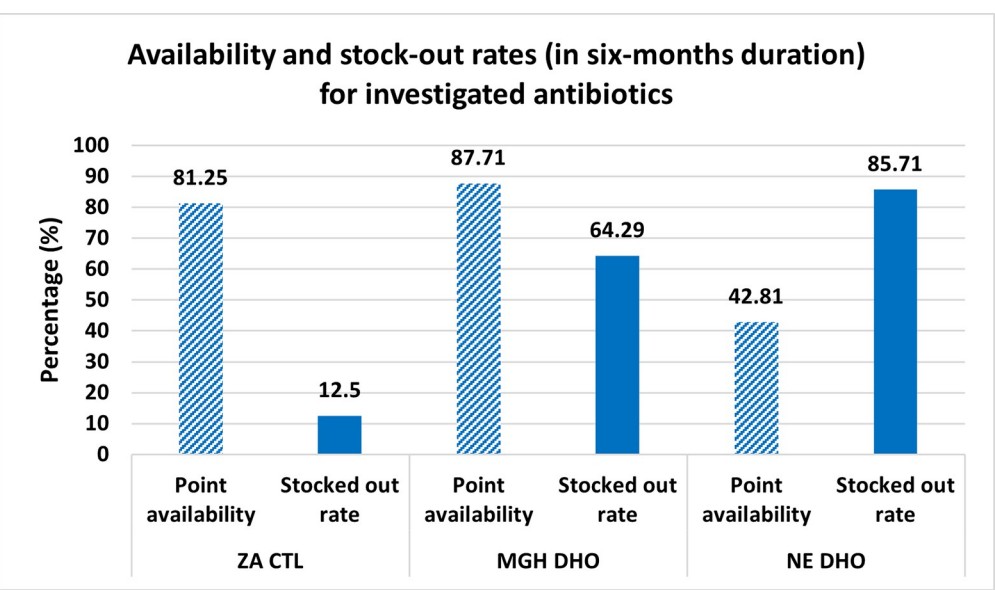

**Fig 1. Percentage of investigated antibiotics which were available on the day of data collection and for which stock-outs were reported in six-month duration.** ZA CTL = Zomba Central Hospital, MGH DHO = Machinga District Hospital, NE DHO = Nsanje District Hospital.

days) for Nsanje, Machinga and Zomba respectively. For Zomba Central Hospital, only erythromycin and ciprofloxacin had stock outs for 28 and five days respectively. Medicines with highest stock out days at Machinga District Hospital were cloxacillin and nalidixic acid, which were not available for the entire six month-duration (**Fig 2**). These were followed by cotrimoxazole (59 days), gentamicin (31 days), ceftriaxone (28 days), benzathine benzylpenicillin (25 days) erythromycin (23 days), doxycycline (19 days) and azithromycin (10 days). Nsanje District Hospital had the highest level of antibiotic stock outs. Azithromycin, cloxacillin and nalidixic acid were not available for the whole six month-duration and were followed by ciprofloxacin (113 days), erythromycin (107 days), benzylpenicillin (98 days), amoxicillin (76 days), flucloxacillin (57 days), gentamicin (40 days), doxycycline (37 days), and metronidazole (17 days).

## Use of antibiotic therapy

Common diagnoses among participants were sepsis (25.7%), pneumonia (19.7%), meningitis (4.9%), cellulitis (3.6%) and peptic ulcers (3.0%). A total of 471 antibiotic medicines were prescribed. The most prescribed antibiotic was ceftriaxone (46.7%, n = 220). This was followed by metronidazole (22.5%, n = 106), benzylpenicillin (13.0%, n = 61), gentamicin (6.4%, n = 30), amoxicillin (4.3%, n = 20), cotrimoxazole (1.5%, n = 7), doxycycline (1.3%, n = 6), flucloxacillin (1.1%, n = 5), erythromycin (0.9%, n = 4), amoxicillin/clavulanic acid (0.4%, n = 2), azithromycin (0.2%, n = 1) and benzathine benzylpenicillin (0.2%, n = 1) (**Fig 3**). Among the participants, 52.3% received single antibiotic therapy while 45.7% received a combination therapy of antibiotics. Common antibiotic combinations were ceftriaxone and metronidazole (17.4%, n = 53); benzyl penicillin and gentamicin (5.3%, n = 16); and amoxicillin and metronidazole (4.3%, n = 13) (**S1 Fig**).

By facilities, ceftriaxone was the most prescribed antibiotic at Zomba (75.6%, n = 132) and Nsanje (32%, n = 48), while benzyl penicillin was the most prescribed antibiotic at Machinga (28.5%, n = 42). In terms of patient diagnosis, ceftriaxone was mostly prescribed for sepsis

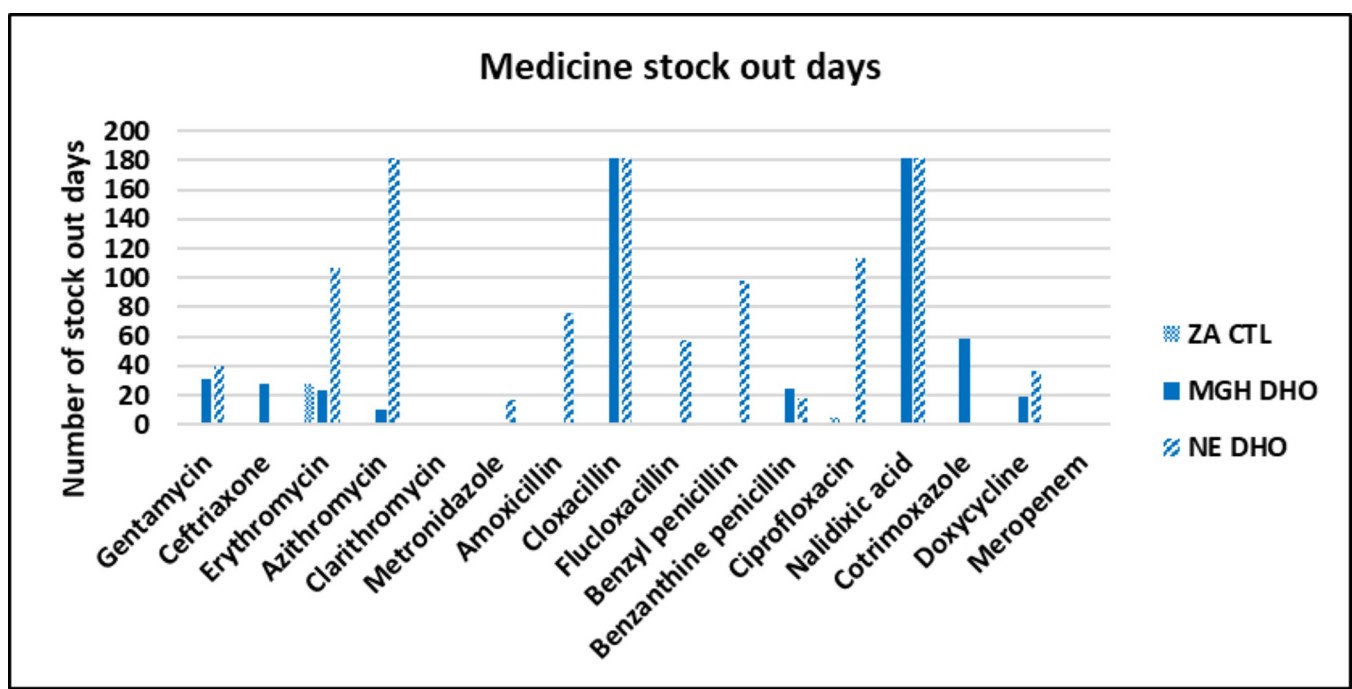

ZA CTL = Zomba Central Hospital, MGH DHO = Machinga District Hospital, NE DHO = Nsanje District Hospital

**Fig 2. Stock out duration for various antibiotic medicines per facility in six-month duration.** ZA CTL = Zomba Central Hospital, MGH DHO = Machinga District Hospital, NE DHO = Nsanje District Hospital.

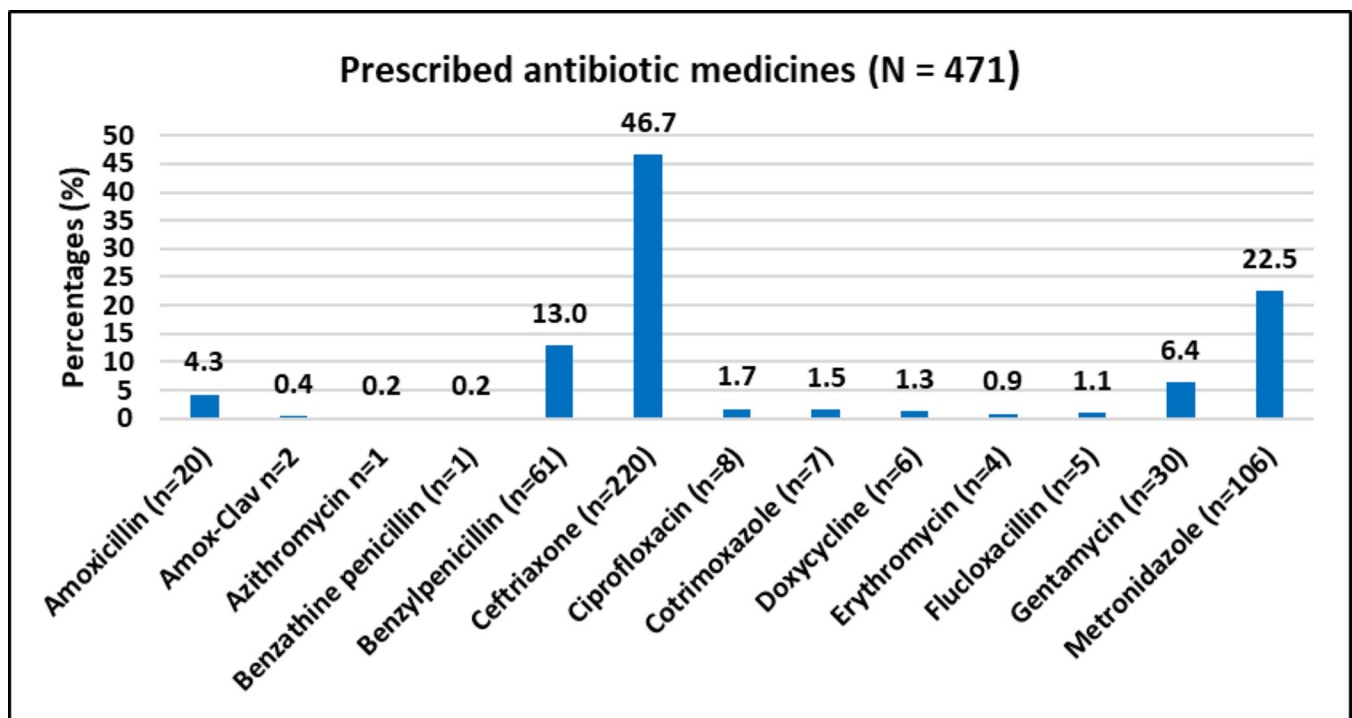

**Fig 3. Prescribed antibiotic therapy in the study.**

**Table 3. Prescribed antibiotics by facility and diagnosis.**

| Name of antibiotic | By facility n, (%) | | | By diagnosis n, (%) | | | | | |
|---|---|---|---|---|---|---|---|---|---|
| | Zomba N = 174 | Machinga N = 147 | Nsanje N = 150 | Sepsis N = 118 | Pneumonia N = 88 | Meningitis N = 20 | Cellulitis N = 21 | Peptic ulcers N = 17 | Others N = 207 |
| Amoxicillin | 5 (2.9) | 2 (1.4) | 13 (8.7) | 5 (4.2) | 1 (1.1) | 0 (0) | 0 (0) | 4 (23.5) | 10 (4.9) |
| Amox- Clav | 2 (1.2) | 0 (0) | 0 (0) | 0 (0) | 2 (2.3) | 0 (0) | 0 (0) | 0 (0) | 0 (0) |
| Azithromycin | 0 (0) | 1 (0.7) | 0 (0) | 0 (0) | 0 (0) | 0 (0) | 0 (0) | 0 (0) | 1 (0.5) |
| Benzathine benzylpenicillin | 0 (0) | 0 (0) | 1 (0.7) | 0 (0) | 0 (0) | 0 (0) | 0 (0) | 0 (0) | 1 (0.5) |
| Benzylpenicillin | 1 (0.6) | 42 (28.5) | 18 (12) | 15 (12.7) | 11 (12.5) | 1 (5) | 3 (14.3) | 4 (11.8) | 29 (14.0) |
| Ceftriaxone | 132 (75.9) | 40 (27.2) | 48 (32) | 60 (50.9) | 50 (56.9) | 14 (70) | 8 (38.1) | 3 (17.7) | 85 (41.1) |
| Ciprofloxacin | 1 (0.6) | 4 (2.7) | 3 (2) | 2 (1.7) | 1 (1.1) | 2 (10) | 0 (0) | 0 (0) | 3 (3.1) |
| Cotrimoxazole | 1 (0.6) | 1 (0.7) | 5 (3.3) | 2 (1.7) | 1 (1.1) | 1 (5) | 1 (4.8) | 0 (0) | 2 (1.0) |
| Doxycycline | 0 (0) | 0 (0) | 6 (4) | 3 (2.5) | 0 (0) | 0 (0) | 0 (0) | 0 (0) | 3 (1.45) |
| Erythromycin | 2 (1.15) | 1 (0.7) | 1 (0.7) | 0 (0) | 0 (0) | 0 (0) | 0 (0) | 0 (0) | 4 (1.9) |
| Flucloxacillin | 1 (0.6) | 0 (0) | 4 (2.7) | 1 (0.9) | 0 (0) | 0 (0) | 3 (14.3) | 0 (0) | 1 (0.5) |
| Gentamycin | 1 (0.6) | 20 (13.6) | 9 (6) | 8 (6.8) | 8 (9.1_ | 2 (10) | 0 (0) | 1 (5.9) | 11 (5.3) |
| Metronidazole | 28 (16.1) | 36 (24.5) | 42 (28) | 22 (18.6) | 14 (15.9) | 0 (0) | 6 (28.6) | 7 (41.2) | 57 (27.5) |

(50.9%, n = 60), pneumonia (56.9%, n = 50), meningitis (70%, n = 14), and cellulitis (38.1%, n = 8). In peptic ulcers, the most prescribed antibiotics were metronidazole (41.2%, n = 8) and amoxicillin (23.5%, n = 4) (**Table 3**).

## Adherence to treatment guidelines

Only 66% (201/304) of cases were assessed for adherence of antibiotic prescribing to treatment guidelines, as the MSTG does not include adequate information on the use of antibiotics for prophylaxis and other uncommon conditions. Adherence to MSTG was associated with health facility, presence of laboratory test results (FBC), spectrum of prescribed antibiotics, and WHO-AWaRe classification of the medicine, $p < 0.005$. Overall, adherence to MSTG was 79.6%, (95% CI, 73.3–84.9%). By facilities, adherence to guidelines at Zomba was 95.9% (95% CI, 89.7–98.9%) while at Nsanje and Machinga was 73.2% (95% CI, 59.7–84.2%) and 54.2% (95% CI, 39.2–68.6%) respectively (**Table 4**). There was high adherence to MSTG for management of meningitis with 93.3% (95% CI, 68–99.8%) of the cases being treated according to the guidelines. The adherence was 90.9% (95% CI, 58.7–99.7%) for cellulitis, 83.3% (95% CI, 71.4–91.7%) for pneumonia, 79.5% (95% CI, 68.8–87.8%) for sepsis, 44% (95% CI, 13.7–78.8%) for peptic ulcers and 71.4% (95% CI, 51.3–86.8%) for miscellaneous conditions such as urinary tract infections, syphilis, and dysentery. There was 93.2% (95% CI, 87.9–96.7%) adherence on the prescribing of broad-spectrum antibiotics while for narrow-spectrum antibiotics adherence was 10% (95% CI, 2.5–44.5%). By AWARE classification, adherence for access antibiotics was 27.3% (95%, 14.9–42.8%), while for watch antibiotics was 98% (95% CI, 92.8–99.7%). Adherence to MSTG was lower for antibiotics that were stocked out (63.8%, 95% CI 48.5–77.3%) than antibiotics that were not stocked out during the study period (84.4%, 95% CI 77.7–89.8%), $p < 0.002$.

## Discussion

This study was conducted at one tertiary (central) hospital and two secondary-level (district) hospitals in southern Malawi. We found that the selected tertiary-level facility had better

**Table 4. Adherence of antibiotic treatment to Malawi Standard Treatment Guidelines.**

| Variable | Characteristic | Total | Adherence to MSTG | | P value[1] |
|---|---|---|---|---|---|
| | | | Yes | No | |
| Age | <30 | 57 | 40 (70.2) | 17 (29.8 | 0.133 |
| | 31–45 | 52 | 41 (78.8) | 11 (21.2) | |
| | 46–64 | 42 | 37 (88.1) | 5 (11.9) | |
| | > 65 | 50 | 42 (84.1) | 8 (16.0) | |
| Sex | Female | 95 | 79 (83.2) | 16 (16.8) | 0.236 |
| | Male | 106 | 81 (76.4) | 25 (23.6) | |
| Facility | ZA CTL | 97 | 93 (95.9) | 4 (4.1) | <0.001 |
| | MGH DHO | 48 | 26 (54.2) | 22 (45.8) | |
| | NE DHO | 56 | 41 (73.2) | 15 (26.8) | |
| Diagnosis | Sepsis | 78 | 62 (79.5) | 16 (20.5) | 0.072 |
| | Pneumonia | 60 | 50 (83.3) | 10 (16.7) | |
| | Meningitis | 15 | 14 (93.3) | 1 (6.7) | |
| | Cellulitis | 11 | 10 (90.9) | 1 (9.1) | |
| | Peptic ulcers | 9 | 4 (44.4) | 5 (55.6) | |
| | Others | 28 | 20 (71.4) | 8 (28.6) | |
| FBC test | Not Done | 81 | 55 (67.9) | 26 (32) | 0.001 |
| | Done | 120 | 105 (87.5) | 15 (12.5) | |
| Diagnosis by system | Cardiovascular | 1 | 1 (100) | 0 (0) | 0.031 |
| | Central Nervous | 16 | 14 (87.5) | 2 (12.5) | |
| | Dermatological | 20 | 17 (85.0) | 3 (15.0) | |
| | Gastrointestinal | 17 | 8 (47.1) | 9 (52.9) | |
| | Immune | 77 | 61 (79.2) | 16 (20.8) | |
| | Respiratory | 66 | 55 (83.3) | 11 (16.7) | |
| | Urogenital | 4 | 4 (100) | 0 (0) | |
| Spectrum | Broad | 148 | 138 (93.2) | 10 (6.8) | <0.001 |
| | Narrow | 10 | 1 (10) | 9 (90) | |
| | Broad + Narrow | 43 | 21 (48.8) | 22 (51.2) | |
| WHO AWARE | Access | 44 | 12 (27.3) | 32 (72.7) | <0.001 |
| | Watch | 97 | 97(98.0) | 2 (2.0) | |
| | Access + Watch | 58 | 51 (87.9) | 7 (12.1) | |

[1]P values are based on chi-square test. Where the cell number was ≤5, Fisher's exact test was used. P values ≤ 0.05 were considered significant.

availability of the required antibiotics compared with the two district hospitals, with one having much worse availability.

Despite the differences in the level of patient care, all public health facilities procure most of their medicines through the Central Medicine Stores Trust (CMST) with few exceptions [35]. During the period of data collection, the central hospitals were allowed to purchase any medicine/pharmaceutical product that was not available at CMST from private pharmaceutical suppliers, unlike district hospitals. Between 2019 and 2023, budget allocation for pharmaceutical purchases has been between 20 and 30 million US$ per year of which approximately 40% is meant for four central hospitals while the rest is shared among the 28 districts [36, 37]. The flexibility by central hospitals to use their budgetary allocation in purchasing from private suppliers and the increased share of the budget best explains the higher availability of antibiotics in central hospitals than in district hospitals.

The availability of antibiotics at Nsanje District Hospital was very poor as compared to Machinga District Hospital. As noted, the point availability was below 50% at Nsanje District

Hospital. In addition, more than 80% of the antibiotics were stocked out within a six-month duration. This could be partially attributed to tropical cyclone Ana and cyclone Dumako which heavily hit the district in January and February 2022 respectively [38]. The disaster resulted in disruption of healthcare delivery including pharmaceutical logistics and supply as most of the roads were rendered impassable. All public health facilities in Malawi receive medical supplies once a month from CMST, as such, the impassable roads meant that there were very few deliveries of normal supply to Nsanje. Moreover, healthcare workers including pharmacy personnel were temporarily re-deployed to various camps where they provided relief healthcare services [38] and thus, affecting quantification and procurement of pharmaceuticals.

The MEML classifies medicines as non-essential, essential, and vital (see legend of Table 1). On the other hand, WHO classifies antibiotic medicines as access, watch, and reserve antibiotics (AWaRe classification) [39]. Most of the antibiotic medicines in MEML are categorized as vital medicines. In our study, only azithromycin and cloxacillin were classified as essential medicines. Both azithromycin and cloxacillin had poor availability at district hospitals but were constantly available at central hospital. By WHO AWaRe categories, the availability of antibiotics did not vary significantly between classes. As noted, 51.8% of access antibiotics were stocked out within six months, and 42.9% of watch antibiotics were stocked out within the same duration. It is most likely that procurement of antibiotics in Malawi does not consider the WHO AWaRe classifications as at the time of the study, few guiding documents aligned the MEML with the WHO AWaRe principles. On the other hand, the latest edition of the WHO model EML was compiled based on the AWaRe classification. This was done with the intentions of improving not just the access to antibiotic therapy but also the quality of antibiotic use, as a way of minimizing antibiotic resistance [40]. The incorporation of the AWARE category in the current 2023 MSTG is a positive development for the future of antibiotic procurement and prescribing in Malawi [41].

The 2015 MSTG provided limited information on the use of antibiotics for prophylaxis and uncommon conditions. The adherence to MSTG for such cases was therefore not assessed in the study. Consistent with other LMICs and previous studies done in Malawi, sepsis and pneumonia were the most common diagnoses in this study [1, 42]. However, the level of diagnostic certainty was low as the diagnoses were all based on clinical presentations without laboratory confirmation [43]. Only FBC was applied in 57% of cases in this study. However, studies have reported a low sensitivity and specificity in the diagnosis of bacterial infections using FBC [44]. As further noted in this study, no sensitivity test or bacterial culture was conducted. This could be one of the reasons for more prescriptions of broad-spectrum antibiotics than narrow-spectrum antibiotics.

Globally, it is recommended that adherence to treatment guidelines for prescribing antibiotics should be >95%. In this study, we found varying rates of adherence to MSTG among the facilities. The adherence to MSTG was assessed based on the records of prescribed antibiotic regimens to patients. Although nurses record in the patient management files when administering the medicines (by indicating the time of administration), it was difficult to confirm retrospectively whether the medicines were administered correctly. Zomba Central Hospital had the highest and optimal level of adherence to guidelines with 95.9% of antibiotics prescribed according to the MSTG. The adherence to guidelines for Zomba Central Hospital was higher than the results of a study conducted at Queen Elizabeth Central Hospital (QECH) in Blantyre district in 2018, which found 84% adherence to antibiotic prescriptions to treatment guidelines [42]. The QECH is one of the four major referral and teaching hospitals in Malawi [45]. Adherence to MSTG was sub-optimal for both Machinga and Nsanje District Hospitals (54.2% and 73.2% respectively).

In this study, we noted that most of the antibiotic therapies that were not in line with the MSTG were combination therapies as compared to single antibiotic therapies. Nevertheless, some of the common antibiotic combinations were rational. For instance, the combination of ceftriaxone with metronidazole is recommended for sepsis if the presumed or known source is intrabdominal. A combination of amoxicillin and metronidazole is also a standard care for the eradication of *Helicobacter pylori* in peptic ulcers [18]. Availability of first-line antibiotics was significantly associated with the level of adherence to MSTG. As noted, although for the two district hospitals, the availability of antibiotics medicines was better in Machinga, the overall adherence to MSTG was poorer as compared to Nsanje District Hospital. This was because Machinga experienced the longest duration for stock out of ceftriaxone, which is the agent of choice for severe sepsis and pneumonia. This could therefore explain why more cases were managed without following guidelines at Machinga District Hospital as compared to Nsanje District Hospital.

The WHO set a target that more than 60% of antibiotics prescribed at the national level should be from the Access category [46]. Although the choice of ceftriaxone is justifiable in most cases according to the MSTG, the overuse of the antibiotic in Malawi is a concern. Ceftriaxone is categorized as a Watch antibiotic by WHO. Thus, the use of this antibiotic needs to be controlled since it has an increased risk of inducing resistance [34]. In addition, cephalosporin antibiotics have been reported to induce cross-resistance with penicillin antibiotics, which are commonly prescribed in primary care in Malawi [11].

## Limitations

This study focused on a few facilities which were only one central and two district hospitals. Primary health center facilities, which serve most of the population in these districts, were not included in the study. The retrospective study design for assessing adherence to guidelines is also another limitation as the data may be biased by missing information in the case management files. In addition, the study did not explore further other possible factors that affected clinician's decisions on the selection of antibiotics.

## Conclusion

The current study revealed that stock out of antibiotic medicines is still a challenge across facilities. The linkage between availability and adherence to standard treatment guidelines suggests that poor availability of medicines may be one of the contributing factors to inappropriate use of antibiotic therapy and consequently increase the risk of antibiotic resistance. Further studies that can validate these findings are necessary for further guidance in antibiotic selection and prescribing policy. In addition, we also recommend the adoption of WHO AWaRe classification to help minimize the overuse of antibiotics with a high risk of antibiotic resistance.

## Supporting information

**S1 Checklist. STROBE statement—checklist of items that should be included in reports of observational studies.**
(DOCX)

**S1 Table. Antibiotic treatment protocols for common bacterial infections in Malawi based on 2015 Malawi Standard Treatment Guidelines.**
(PDF)

**S2 Table. Data collection forms.**
(PDF)

**S1 Fig. Showing combination of antibiotics prescribed.**
(PNG)

## Acknowledgments

We are grateful to the pharmacists, nursing, and data officers of the respective hospitals for supporting us during data collection.

## Author Contributions

**Conceptualization:** Francis Kachidza Chiumia, Adamson Sinjani Muula, Elizabeth Kampira, Felix Khuluza.

**Data curation:** Francis Kachidza Chiumia, Frider Chimimba, Happy Magwaza Nyirongo, Elizabeth Kampira, Felix Khuluza.

**Formal analysis:** Francis Kachidza Chiumia, Adamson Sinjani Muula, Frider Chimimba, Happy Magwaza Nyirongo, Elizabeth Kampira, Felix Khuluza.

**Funding acquisition:** Adamson Sinjani Muula, Felix Khuluza.

**Investigation:** Francis Kachidza Chiumia, Happy Magwaza Nyirongo, Felix Khuluza.

**Methodology:** Francis Kachidza Chiumia, Adamson Sinjani Muula, Frider Chimimba, Happy Magwaza Nyirongo, Elizabeth Kampira, Felix Khuluza.

**Project administration:** Francis Kachidza Chiumia, Felix Khuluza.

**Resources:** Happy Magwaza Nyirongo, Felix Khuluza.

**Software:** Felix Khuluza.

**Supervision:** Adamson Sinjani Muula, Frider Chimimba, Elizabeth Kampira, Felix Khuluza.

**Validation:** Adamson Sinjani Muula, Felix Khuluza.

**Writing – original draft:** Francis Kachidza Chiumia, Adamson Sinjani Muula, Frider Chimimba, Happy Magwaza Nyirongo, Felix Khuluza.

**Writing – review & editing:** Francis Kachidza Chiumia, Adamson Sinjani Muula, Frider Chimimba, Elizabeth Kampira, Felix Khuluza.

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
