## [Decision Letter · Decision Letter 0]

25 Aug 2023

PONE-D-23-20272Impact of antibiotic medicines availability on adherence to standard treatment guidelines among hospitalized adult patients in Southern MalawiPLOS ONE

Dear Dr. Khuluza,

Thank you for submitting your manuscript to PLOS ONE. After careful consideration, we feel that it has merit but does not fully meet PLOS ONE’s publication criteria as it currently stands. Therefore, we invite you to submit a revised version of the manuscript that addresses the points raised during the review process.

We look forward to receiving your revised manuscript.

Kind regards,

Balew Arega Negatie, Msc,MD

Academic Editor

PLOS ONE

Reviewers' comments:

Reviewer's Responses to Questions

**Comments to the Author**

1. Is the manuscript technically sound, and do the data support the conclusions?

Reviewer #1: No

Reviewer #2: Yes

2. Has the statistical analysis been performed appropriately and rigorously? 

Reviewer #1: I Don't Know

Reviewer #2: Yes

3. Have the authors made all data underlying the findings in their manuscript fully available?

Reviewer #1: No

Reviewer #2: Yes

4. Is the manuscript presented in an intelligible fashion and written in standard English?

Reviewer #1: Yes

Reviewer #2: Yes

5. Review Comments to the Author

Reviewer #1: This an interesting read on adherence to antibiotic therapy and its possible potential contribution to the development of antimicrobial resistance in a sub-Saharan African country where the problem is most acute. The authors look into availability of antibiotics and adherence to existing guidelines.

However, there are a number of major issues to be addressed before this write-up can be considered for publication.

Some of the minor issues (there are many more and they can’t all be addressed here):

1 - Expressions such as “antibiotic medications”, “medicines” are very confusing and should be replaced to display a focus on antibiotic therapy.

2 - Key words: there are too many of them, some including acronyms. They contain both “antibiotic resistance” AND “antimicrobial resistance” which to me, have the same meaning. Besides, “antibiotic resistance” extensively used in the manuscript is confusing. I guess authors are referring to “resistance to antibiotics”.

3 - I do not think the tool used for data collection was a simple “questionnaire”.

4 – The background section is way too long and elaborates a lot on issues not necessarily related to the problem the authors are attempting to address.

5 – The methodology section does not provide details of how “data on medicine stock-outs” were obtained.

6 – if the study was carried-out in two secondary hospitals, how did you analyse Meropenem and Clarithromycin which are found only in tertiary hospitals?

Some of the major issues:

1 – The insight into the “Malawi Standard Treatment Guidelines” which serves as a basis to assess adherence to treatment and is succinctly evoked in the background section (it should actually appear in the methodology section) needs to be more elaborate. All the reader has access to is a list of antibiotics and no one understand how this was arrived at. Providing a deeper insight into this would help the reader understand the data provided into table 1 (classification of antibiotics) and the reason why formulations such as Benzathin Penicillin, Benzyl Penicilin (these two are actually the same), Amoxicillin, ampicillin, Flucloxacillin, etc. are still in use in their settings though all available data point to their extensive resistance pattern for decades now. And the authors state that “the Malawi antimicrobial resistance strategy was established in 2017 and aimed at achieving 100% optimization of the use of antibiotics by 2022”.

2 – If the aim of this study was to analyse the availability of “selected” antibiotics, what proportion of antibiotics used was selected and what were the criteria for selection. Why select in the first place rather than performing a more complete analysis?

3 - Though adherence to treatment could easily be connected to availability of antibiotics, the authors never attempted to correlate them. Adherence to an antibiotic regiment could be influenced by a wide range of other factors such as socio-economic status and capability to acquire available antibiotics, tolerability of treatment, etc.

4 – The authors assessed point-of-care availability of antimicrobials and ignored a VERY frequent practice in many countries: an antibiotic not available at the point-of-care if often acquired by patients and their relatives in other delivery point such as town pharmacies and road side vendors. This completely biases the availability of antibiotics as reported in this write-up.

5 – It would have been easy to collect additional data on the suitability of the antibiotic regimens applied to patients (choice of antibiotics, duration of treatment, etc.) irrespective of wither the treatment was proposed based on laboratory workup or in the spirit of a probabilistic treatment.

Reviewer #2: PONE-D-23-20272 Impact of antibiotic medicines availability on adherence to standard treatment guidelines among hospitalized adult patients in Southern Malawi

I enjoyed reading this manuscript and congratulate the authors on their work. It is important that more research on topics relevant to low- and middle-income countries is carried out and published, especially by researchers from those countries. The present manuscript, by researchers from the Kamuzu University of Health Sciences in Malawi, is a very encouraging example. It addresses very important questions: the authors provide evidence that there is a need to consider antibiotic availability as one of the key issues to be addressed in order to curb antibiotic resistance.

I have only minor suggestions for modifications:

1) Abstract, line 72: the expression “characteristics” of antibiotics may not be clear to the reader. I suggest to explain here which type of “characteristics” are meant.

2) Background: This section is very good, both in content and in wording.

3) Methods, line 142: state how many districts exist in Southern Malawi.

4) Line 151-152: I do not understand the sample size calculation. It was probably based on some study hypothesis, an expected difference between two groups, or similar. If possible, please explain.

5) Line 160: Please explain to whom the structured questionnaire was administered, and how. Include the complete questionnaire as Supplementary Materials.

6) Line 161: The authors correctly state that they recorded which medicines were prescribed. Maybe in the Discussion section, they can briefly discuss whether it was likely that these medicines were indeed administrated to the patients as prescribed.

7) Line 191: add a citation for WHO AWaRe.

8) Table 1: maybe re-check: is meropenem “Watch”, or “Reserve”?

9) Table 2: the two different headings for column 3 (and also column 4) and are confusing. Consider using only “Median (IQR)” in the first line, and then add an additional line after line 2, and give there the heading for all subsequent data as “N (%)”

10) Table 2: I suggest to delete the lines referring to “Classification by MELM”. All investigated antibiotics belong to the same category anyway (“vital”), and this is better stated in the text.

11) Line 258: For clarity, I suggest to add “in a six month period”.

12) Table 3 and text: as stated in lines 263 and 209, a total of 471 antibiotic medicines were prescribed to 304 patients. This means that about half of the patients received combinations of antibiotics. The manuscript should include data which combinations were used, e.g.:

a) Table 3 may show two additional columns, giving for each antibiotic the number of “monotherapies" and “combination therapies” for which this antibiotic was used.

b) The Results text should state the most frequently observed combinations of antibiotics. The Discussion may address whether these combinations followed MSTG and the principles of rational antibiotic treatment.

13) Line 283: “characteristics”: see remark No. 1 above.

14) Line 312: Consider rewording this sentence, e.g. to “The availability of antibiotics at Nsanje District Hospital was very poor”.

15) Discussion: I suggest to delete lines 324-341, in order to make the Discussion more concise, and to instead add a remark “(see legend of Table 1)” in line 343.

16) Line 380: I appreciate that the authors correctly address the important issue of ceftriaxone overuse.

17) Fig. 1A: I suggest to omit the three columns showing “not available”. They give only redundant information.

18) Fig. 2A: The title “Antibiotic stock-out in 6 month duration” is not clear enough. Consider to write either in the title or in the legend something like “Percentage of the investigated antibiotics for which stock-outs were reported in a 6 month period”.

6. PLOS authors have the option to publish the peer review history of their article (what does this mean?). If published, this will include your full peer review and any attached files.

Reviewer #1: **Yes: **Mefire Alain Chichom

Reviewer #2: No

---

## [Author Response · Author response to Decision Letter 0]

4 Sep 2023

Section A: Journal requirements:

Response: We have reworked and followed PLOS guidelines as advised

Response: we have made all the data for this study available in the manuscript as well as supplementary materials.

Response: We have deleted the ethics statement at the end of the manuscript

Section B: Responses to reviewer #1

Reviewer #1: This an interesting read on adherence to antibiotic therapy and its possible potential contribution to the development of antimicrobial resistance in a sub-Saharan African country where the problem is most acute. The authors look into availability of antibiotics and adherence to existing guidelines.

However, there are a number of major issues to be addressed before this write-up can be considered for publication.

Some of the minor issues (there are many more and they can’t all be addressed here):

1 - Expressions such as “antibiotic medications”, “medicines” are very confusing and should be replaced to display a focus on antibiotic therapy.

Response: We have revised accordingly where ‘antibiotic medicines’ was intended to mean ‘antibiotic therapy’ throughout the manuscript

2 - Key words: there are too many of them, some including acronyms. They contain both “antibiotic resistance” AND “antimicrobial resistance” which to me, have the same meaning. Besides, “antibiotic resistance” extensively used in the manuscript is confusing. I guess authors are referring to “resistance to antibiotics”.

Response: We have reduced the number of key words (line 81-82). With reference to the World Health Organization and other sources in literature, the term ‘antibiotic resistance’ is universally used and has the same meaning as ‘resistance to antibiotics’. 

3 - I do not think the tool used for data collection was a simple “questionnaire”.

Response: Medicine availability and patient clinical data were abstracted from stock cards and patient management files respectively and recorded in separate data collection forms by investigators. We have revised the section (line 163-165) and provided the data collection forms as S2 Table. 

4 – The background section is way too long and elaborates a lot on issues not necessarily related to the problem the authors are attempting to address.

Response: We have removed irrelevant sections to make the background section more concise. 

5 – The methodology section does not provide details of how “data on medicine stock-outs” were obtained.

Response: We have provided further details in the methodology section (line 165-175). Stock cards are maintained on shelves in the pharmacy and contain information such as quantity of medicines remaining on shelf and days which the medicine was not available on shelf. This data was collected using a data collection form by investigators (line 165). 

6 – if the study was carried-out in two secondary hospitals, how did you analyse Meropenem and Clarithromycin which are found only in tertiary hospitals?

Response: As stated in line 129-131, the study also included one tertiary hospital where the availability of Meropenem and Clarithromycin were assessed. For the district hospitals, we did not assess the availability of these two medicines as described in the methods section on line 145-149. 

Some of the major issues:

1 – The insight into the “Malawi Standard Treatment Guidelines” which serves as a basis to assess adherence to treatment and is succinctly evoked in the background section (it should actually appear in the methodology section) needs to be more elaborate. All the reader has access to is a list of antibiotics and no one understand how this was arrived at. Providing a deeper insight into this would help the reader understand the data provided into table 1 (classification of antibiotics) and the reason why formulations such as Benzathin Penicillin, Benzyl Penicilin (these two are actually the same), Amoxicillin, ampicillin, Flucloxacillin, etc. are still in use in their settings though all available data point to their extensive resistance pattern for decades now. And the authors state that “the Malawi antimicrobial resistance strategy was established in 2017 and aimed at achieving 100% optimization of the use of antibiotics by 2022”.

Response: We purposively selected first line antibiotics for common bacterial infections in Malawi. We have provided more details on the common bacterial infections in Malawi and the treatment protocols stated in the Malawi Standard Treatment Guidelines (line 134-144). The treatment protocols have been summarized as S1 Table. As noted, most of the recommended antibiotics in Malawi are penicillin antibiotics (due to relatively lower cost). Benzylpenicillin (Penicillin G) and Benzanthine benzylPenicillin (benzanthine penicillin G) are two different medicines. In comparison, benzathine penicillin has a slower rate of absorption, higher stability and usually administered as a single dose mostly for sexually transmitted infections. We have provided the reasons for including both medicines in our study in line 149-154. As highlighted in the manuscript (reference provided), the Malawi antimicrobial strategy of 2017 aimed to achieve 100% optimization for use of antibiotics by improving adherence to these treatment protocols. 

2 – If the aim of this study was to analyse the availability of “selected” antibiotics, what proportion of antibiotics used was selected and what were the criteria for selection. Why select in the first place rather than performing a more complete analysis?

Response: The list represented 45.7% of the antibiotics in adult formulation as stated in line 144-145. The selected medicines are first line recommended therapy for common bacterial infections as described in lines 139-141, and S1 Table. 

3 - Though adherence to treatment could easily be connected to availability of antibiotics, the authors never attempted to correlate them. Adherence to an antibiotic regiment could be influenced by a wide range of other factors such as socio-economic status and capability to acquire available antibiotics, tolerability of treatment, etc.

Reference: In the study, we used chi-square test to assess the association between availability of antibiotics (or other variables such as health facility and availability of lab test) and adherence to treatment guidelines (Table 4). The influence of factors such as socio-economic status was not assessed because healthcare services in public facilities in Malawi are 100% free for everyone. Therefore, the socio-economic status of the patients could not influence the prescribing of therapy or adherence to prescribed therapy. 

4 – The authors assessed point-of-care availability of antimicrobials and ignored a VERY frequent practice in many countries: an antibiotic not available at the point-of-care if often acquired by patients and their relatives in other delivery point such as town pharmacies and road side vendors. This completely biases the availability of antibiotics as reported in this write-up.

Response: In Malawi, most people live on less than $2.15 per day and do not have health insurance. Affordability is the major reason why they prefer to go to public health facilities to receive free medical services. Although few people would buy medicines from private pharmacies, this practice is rare for hospitalized patients. Normally, the pharmacists inform clinicians about available medicines during morning reports, so that any available alternative medicines are instead prescribed. In addition, there are limited private pharmacies in the districts where the study was conducted. For instance, at the time of the study, Nsanje district had no pharmacy but only medicine stores, which are not allowed to stock prescription only medicines such as antibiotics. 

5 – It would have been easy to collect additional data on the suitability of the antibiotic regimens applied to patients (choice of antibiotics, duration of treatment, etc.) irrespective of wither the treatment was proposed based on laboratory workup or in the spirit of a probabilistic treatment.

Response: As stated in line 177-179, the suitability of the choice and duration of antibiotic treatment was assessed based on comparison with what is stated in the standard treatment guidelines. This was done on the assumption that the diagnosis was correct, regardless of whether it was confirmed by laboratory test results or not.

Response to reviewer #2

Reviewer #2: PONE-D-23-20272 Impact of antibiotic medicines availability on adherence to standard treatment guidelines among hospitalized adult patients in Southern Malawi

I enjoyed reading this manuscript and congratulate the authors on their work. It is important that more research on topics relevant to low- and middle-income countries is carried out and published, especially by researchers from those countries. The present manuscript, by researchers from the Kamuzu University of Health Sciences in Malawi, is a very encouraging example. It addresses very important questions: the authors provide evidence that there is a need to consider antibiotic availability as one of the key issues to be addressed in order to curb antibiotic resistance.

I have only minor suggestions for modifications:

1) Abstract, line 72: the expression “characteristics” of antibiotics may not be clear to the reader. I suggest to explain here which type of “characteristics” are meant.

Response: The characteristics is based on spectrum of activity and WHO AWaRe classification of the antibiotic medicine. This has been explained in line 72-73.

2) Background: This section is very good, both in content and in wording.

Response: Thank you very much for the positive comment

3) Methods, line 142: state how many districts exist in Southern Malawi.

Response: We have indicated the total number of districts in Southern Malawi, among which our study districts were selected (line 132).

4) Line 151-152: I do not understand the sample size calculation. It was probably based on some study hypothesis, an expected difference between two groups, or similar. If possible, please explain.

Response: This study was part of another study which assessed clinical outcomes among these participants. Parameters used for calculation of sample size have been stated in line 156-158. 

5) Line 160: Please explain to whom the structured questionnaire was administered, and how. Include the complete questionnaire as Supplementary Materials.

Response: We have provided more clarity on the data collection procedure. In brief, investigators reviewed stock cards and patient management files (line 163-179). All the information was recorded in data collection forms by the investigators. The data collection forms have been provided as S 2 file. 

Response:

6) Line 161: The authors correctly state that they recorded which medicines were prescribed. Maybe in the Discussion section, they can briefly discuss whether it was likely that these medicines were indeed administrated to the patients as prescribed.

Response: This has been discussed accordingly in line 374-377. 

7) Line 191: add a citation for WHO AWaRe.

Response: Citation for the WHO AWaRe has been added (line 207)

8) Table 1: maybe re-check: is meropenem “Watch”, or “Reserve”?

Response: We have cross-checked the classification of Meropenem. Although in Malawi (and other countries) Meropenem is used as on of the last resort antibiotics, WHO classifies it as a Watch antibiotic. However, its combination with beta-lactamase inhibitor such as vaborbactam is classified as a reserve antibiotic. 

9) Table 2: the two different headings for column 3 (and also column 4) and are confusing. Consider using only “Median (IQR)” in the first line, and then add an additional line after line 2, and give there the heading for all subsequent data as “N (%)”

Response: We have revised the heading. For “Median (IQR)”, we have provided details as table footnote (Table 2)

10) Table 2: I suggest to delete the lines referring to “Classification by MELM”. All investigated antibiotics belong to the same category anyway (“vital”), and this is better stated in the text.

Response: The rows for MELM classification have been deleted in Table 2

11) Line 258: For clarity, I suggest to add “in a six month period”.

Response: We have added ‘in six-month duration’ for clarity as suggested (line 273) 

12) Table 3 and text: as stated in lines 263 and 209, a total of 471 antibiotic medicines were prescribed to 304 patients. This means that about half of the patients received combinations of antibiotics. The manuscript should include data which combinations were used, e.g.:

a) Table 3 may show two additional columns, giving for each antibiotic the number of “monotherapies" and “combination therapies” for which this antibiotic was used.

b) The Results text should state the most frequently observed combinations of antibiotics. The Discussion may address whether these combinations followed MSTG and the principles of rational antibiotic treatment.

Response: Data for single and combination antibiotic therapies have been provided as S1 figure. We have described the most frequently observed combinations in the result section (line 283-287) and discussed whether the combinations were in line with MSTG or not in line 384-389. 

13) Line 283: “characteristics”: see remark No. 1 above.

Response: As done in response 1, we have also similarly explained the characteristics in line 302-303. 

14) Line 312: Consider rewording this sentence, e.g. to “The availability of antibiotics at Nsanje District Hospital was very poor”.

Response: We have revised the sentence as suggested (line 335-336). 

15) Discussion: I suggest to delete lines 324-341, in order to make the Discussion more concise, and to instead add a remark “(see legend of Table 1)” in line 343.

Response: We agree with the reviewer’s observation, and we have deleted the section as suggested. For details on the MEML classification, we have referred to legend of Table 1(line 346). 

16) Line 380: I appreciate that the authors correctly address the important issue of ceftriaxone overuse.

Response: Thank you for the positive comment

17) Fig. 1A: I suggest to omit the three columns showing “not available”. They give only redundant information.

Response: We have omitted the columns for ‘not available’ in figure 1A and ‘not-stocked out’ in figure 1B. Figures 1A and 1B have been combined as Figure 1 for proper illustration as the number of columns has been reduced. 

18) Fig. 2A: The title “Antibiotic stock-out in 6 month duration” is not clear enough. Consider to write either in the title or in the legend something like “Percentage of the investigated antibiotics for which stock-outs were reported in a 6 month period”.

Response: The title mentioned was for figure 1B (which has been revised as in response 17) and we have revised the title for the updated figure as suggested (line 256-257).

---

## [Decision Letter · Decision Letter 1]

10 Oct 2023

PONE-D-23-20272R1Impact of antibiotic medicines availability on adherence to standard treatment guidelines among hospitalized adult patients in Southern MalawiPLOS ONE

Dear Dr. Khuluza,

Thank you for submitting your manuscript to PLOS ONE. After careful consideration, we feel that it has merit but does not fully meet PLOS ONE’s publication criteria as it currently stands. Therefore, we invite you to submit a revised version of the manuscript that addresses the points raised during the review process.

Please submit your revised manuscript by Nov 24 2023 11:59PM. If you will need more time than this to complete your revisions, please reply to this message or contact the journal office at plosone@plos.org. Please include the following items when submitting your revised manuscript:A rebuttal letter that responds to each point raised by the academic editor and reviewer(s). You should upload this letter as a separate file labeled 'Response to Reviewers'.A marked-up copy of your manuscript that highlights changes made to the original version. You should upload this as a separate file labeled 'Revised Manuscript with Track Changes'.An unmarked version of your revised paper without tracked changes. You should upload this as a separate file labeled 'Manuscript'.We look forward to receiving your revised manuscript.

Kind regards,

Balew Arega Negatie, Msc,MD

Academic Editor

PLOS ONE

Journal Requirements:

Additional Editor Comments (if provided):

Dear Author

See my minor comments / changes/ highlighted in the main text .

Reviewers' comments:

Reviewer's Responses to Questions

**Comments to the Author**

1. If the authors have adequately addressed your comments raised in a previous round of review and you feel that this manuscript is now acceptable for publication, you may indicate that here to bypass the “Comments to the Author” section, enter your conflict of interest statement in the “Confidential to Editor” section, and submit your "Accept" recommendation.

Reviewer #1: All comments have been addressed

Reviewer #2: (No Response)

2. Is the manuscript technically sound, and do the data support the conclusions?

Reviewer #1: Yes

Reviewer #2: Yes

3. Has the statistical analysis been performed appropriately and rigorously? 

Reviewer #1: Yes

Reviewer #2: Yes

4. Have the authors made all data underlying the findings in their manuscript fully available?

Reviewer #1: Yes

Reviewer #2: Yes

5. Is the manuscript presented in an intelligible fashion and written in standard English?

Reviewer #1: Yes

Reviewer #2: Yes

6. Review Comments to the Author

Reviewer #1: (No Response)

Reviewer #2: All my comments to the previous version of the manuscript have been correctly addressed.

As far as I can see, also the comments of reviewer #1 have been correctly addressed. In two points, I disagree with previous remarks by reviewer #1 (and agree with the authors):

1) Benzathine benzylpenicillin, benzylpenicillin, amoxicillin, ampicillin and flucloxacillin are extremely important antibiotics, included in the most recent version of the WHO Essential List of Medicines (2023), and are certainly not obsolete.

2) Benzathine benzylpenicillin and benzylpenicillin are two different medicines.

Therefore, I believe that the previous and revised manuscript are correct in these points. Just to avoid confusion, I suggest that the authors use the complete name “benzathine benzylpenicillin” throughout the manuscript, and do not shorten it to “benzathine penicillin” as they have done in a few places. A correct alternative name for benzathine benzylpenicillin is “benzathine penicillin G”, but not “benzathine penicillin”.

In the Supplementary Material for their revised manuscript, the authors have now included S1 Figure to show the frequency of the use of combination therapies. The figure is fine in principle, but needs correction of a serious technical problem: apparently, half of the names for antibiotics or antibiotic combinations on the left side got lost. E.g. in their revised manuscript, lines 285-286, the authors state: “ceftriaxone and metronidazole (17.4%, n=53); benzyl penicillin and gentamicin (5.3%, n=16)”. However, in S1 Figure, while the columns for 17.4% and 5.3% can be found, but they are not labeled with the names of the respective antibiotic combinations.

7. PLOS authors have the option to publish the peer review history of their article (what does this mean?). If published, this will include your full peer review and any attached files.

Reviewer #1: No

Reviewer #2: No

---

## [Author Response · Author response to Decision Letter 1]

13 Oct 2023

Thanks for your e-mail of 11 October 2023, and for the reviewer comments. We have revised the manuscript, considering all comments by the reviewers. We would like to respond to the reviewer comments as follows:

Section A: Journal requirements:

Response: We have reviewed the reference list and have changed the citation to conform to PLOS guidelines

Section B: Responses to reviewer #1

2. Reviewer #1: See my minor comments / changes/ highlighted in the main text .

Response: We have revised accordingly and replaced the word “Impact” with “effect” in the title as suggested. 

We have also made all necessary editing requested throughout the document.

On the issue of benzathine penicillin, we have corrected all to “benzathine benzylpenicillin” as suggested by the reviewer

3. In the Supplementary Material for their revised manuscript, the authors have now included S1 Figure to show the frequency of the use of combination therapies. The figure is fine in principle, but needs correction of a serious technical problem: apparently, half of the names for antibiotics or antibiotic combinations on the left side got lost. E.g. in their revised manuscript, lines 285-286, the authors state: “ceftriaxone and metronidazole (17.4%, n=53); benzyl penicillin and gentamicin (5.3%, n=16)”. However, in S1 Figure, while the columns for 17.4% and 5.3% can be found, but they are not labeled with the names of the respective antibiotic combinations.

We have formatted S1 Figure to show the hidden axis labels (S1 Figure revised)

---

## [Editor Report · Decision Letter 2]

16 Oct 2023

Effect of antibiotic medicines availability on adherence to standard treatment guidelines among hospitalized adult patients in Southern Malawi

PONE-D-23-20272R2

Dear Dr.Francis Kachidza 

We’re pleased to inform you that your manuscript has been judged scientifically suitable for publication and will be formally accepted for publication once it meets all outstanding technical requirements.

Kind regards,

Balew Arega Negatie, Msc,MD

Academic Editor

PLOS ONE
---

## [Editor Report · Acceptance letter]

20 Oct 2023

PONE-D-23-20272R2 

Effect of antibiotic medicines availability on adherence to standard treatment guidelines among hospitalized adult patients in Southern Malawi 

Dear Dr. Khuluza:

I'm pleased to inform you that your manuscript has been deemed suitable for publication in PLOS ONE. Congratulations! Your manuscript is now with our production department. 

Kind regards, 

on behalf of

Dr. Balew Arega Negatie 

Academic Editor

PLOS ONE